# Maize–Soybean Rotation and Intercropping Increase Maize Yield by Influencing the Structure and Function of Rhizosphere Soil Fungal Communities

**DOI:** 10.3390/microorganisms12081620

**Published:** 2024-08-08

**Authors:** Liqiang Zhang, Yuhan Yang, Zehang Zhao, Yudi Feng, Baoyin Bate, Hongyu Wang, Qiuzhu Li, Jinhu Cui

**Affiliations:** College of Plant Science, Jilin University, Changchun 130012, China; lqzhang23@mails.jlu.edu.cn (L.Z.); wakk47640887@163.com (Y.Y.); 18846915612@163.com (Z.Z.); fengyudi1@163.com (Y.F.); bybt23@mails.jlu.edu.cn (B.B.); hong_yu@jlu.edu.cn (H.W.)

**Keywords:** continuous maize cropping, maize–soybean rotation, maize–soybean intercropping, fungal community diversity, fungal community function, soil-borne diseases

## Abstract

Soil-borne diseases are exacerbated by continuous cropping and negatively impact maize health and yields. We conducted a long-term (11-year) field experiment in the black soil region of Northeast China to analyze the effects of different cropping systems on maize yield and rhizosphere soil fungal community structure and function. The experiment included three cropping systems: continuous maize cropping (CMC), maize–soybean rotation (MSR), and maize–soybean intercropping (MSI). MSI and MSR resulted in a 3.30–16.26% lower ear height coefficient and a 7.43–12.37% higher maize yield compared to CMC. The richness and diversity of rhizosphere soil fungi were 7.75–20.26% lower in MSI and MSR than in CMC. The relative abundances of *Tausonia* and *Mortierella* were associated with increased maize yield, whereas the relative abundance of *Solicoccozyma* was associated with decreased maize yield. MSI and MSR had higher proportions of wood saprotrophs and lower proportions of plant pathogens than CMC. Furthermore, our findings indicate that crop rotation is more effective than intercropping for enhancing maize yield and mitigating soil-borne diseases in the black soil zone of Northeast China. This study offers valuable insights for the development of sustainable agroecosystems.

## 1. Introduction

Northeast China, as one of the four major black soil areas worldwide, has become an important grain production area [1]. The grain production capacity of the northeastern region has increased in recent years; however, continuous cropping still dominates [2]. Continuous cropping is unsustainable and has led to soil degradation in the region [3], limiting crop yields. It is vital to find sustainable intensification methods that will ensure food security at a lower cost to resources and the environment [4]. Systems that incorporate crop diversity could solve the above problems.

In recent years, high-throughput sequencing technology has opened up a new era of environmental microbial research. This technology allows us to obtain a large amount of genetic information without the need for culturing samples [5]. Microbial diversity and function are essential for maintaining soil quality and ecosystems [6]. As soil microbial systems are large and complex, they play a key role in the decomposition of soil organic matter, the cycling of important elements such as C and N, the degradation of soil pollutants, and the formation and stabilization of soil structure [7]. In a study comparing maize–soybean rotation with continuous maize cropping, it was found that the soil under maize–soybean rotation had a higher relative abundance of the *nifB* gene, which encodes a nitrogen-fixing enzyme, higher bacterial diversity, and altered bacterial community structure [8]. Specifically, there was a higher relative abundance of Acidobacteria at the phylum level, and a higher and lower relative abundance of *Rhizobiaceae* and *Sphingomonadaceae*, respectively, at the family level [9]. A study comparing maize–soybean intercropping with maize monocropping found that intercropping resulted in a higher number of soil bacteria and actinomycetes, and a higher relative abundance of beneficial bacteria such as Proteobacteria and Acidobacteria, which are dominant soil bacteria [10].

A meta-analysis on the effect of crop rotation on soil biological properties at the global scale found that crop rotation significantly increased soil microbial biomass carbon and nitrogen in areas with a mean annual temperature (MAT) ≥ 8 °C and increased the Shannon diversity of soil bacterial communities in colder areas (MAT < 8 °C) [11]. For different soil types, rotation performs better in coarse or medium textured soil, with medium levels of initial SOC (7–10 g ka^−1^) and low TN (<1.2 g kg^−1^) [12]. Furthermore, a crop rotation increased the Shannon diversity by 6.55% in a region with a MAT of 15 °C [13]. In terms of pests and diseases, one study found that intercropping reduced nematode damage to the target crop by 40% and reduced disease incidence by 55% [14].

Based on the above literature, both intercropping and crop rotation with maize and soybean can increase yields by improving biological characteristics compared to continuous maize cropping. However, these studies focused on microbiological communities. For example, Li et al. (2023) found that crop rotation and intercropping plantings altered bacterial and fungal community composition and improved the abundance of potentially plant-beneficial fungi, such as *Mortierella*, while reducing the abundance of potentially plant-pathogenic fungi, such as *Fusarium* [15]. There are few reports on the role of fungal community structure and function in crop growth and development. We hypothesized that cropping patterns may lead to changes in soil fungal communities and that such changes would inevitably affect crop agronomic traits and yields. To test this hypothesis, we sampled maize rhizosphere soils from an 11-year field experiment that included continuous maize cropping, maize–soybean rotation, and maize–soybean intercropping. We characterized fungal community structure and function using high-throughput sequencing. The objectives of this study were (1) to compare fungal community structure and function in maize rhizosphere soil under the three cropping systems and (2) clarify the relationships between the rhizosphere soil fungal community and maize agronomic traits and yield. This study provides a theoretical basis for promoting the sustainable development of maize cultivation.

## 2. Materials and Methods

### 2.1. Study Area

A long-term field experiment was established in May 2012 in Sijian Village, Chengxi Town, Changchun City, central Jilin Province. The study site was located at the Agricultural Experimental Base of the College of Plant Science, Jilin University (125°26.83′ E, 43°90.77′ N). The mean monthly rainfall and mean air temperature during the crop-growing season are shown in Figure 1. There is a frost-free period of 140 days and a mean annual effective cumulative temperature of 3442.3 °C. The soil is black calcium soil. Before the experiment (25 April 2012), the soil (0–20-cm layer) had the following characteristics: pH, 5.64; organic matter content, 1.89%; total nitrogen content, 1.05 g kg^−1^; total phosphorus content, 0.35 g kg^−1^; total potassium content, 9.75 g kg^−1^; available nitrogen content, 145.82 mg kg^−1^; available phosphorus content, 23.65 mg kg^−1^; available potassium content, 160.03 mg kg^−1^; soil capacity, 1.13 g cm^3^; soil temperature, 14.8 °C; and relative soil water content, 75.83%.

### 2.2. Experimental Design

The maize variety Fumin 985 (128 days growth period, late-maturing, compact variety, 2012–2023) and the soybean variety Changnong 39 (129 days growth period, medium maturity branched variety with an unlimited podding habit, 2012–2023) were used. The experiment was designed as a large-area production trial with three cropping systems, namely, continuous maize cropping (CMC), maize–soybean rotation (maize: 20 rows, soybean: 20 rows; MSR), and maize–soybean intercropping (maize: 20 rows, soybean: 20 rows; MSI). Before the experiment, the site was planted with maize. Four replicate plots were established for MSI and MSR, and one plot for CMC. Soybean was planted at a density of 200,000 plants per hectare and maize at 75,000 plants per hectare (Figure 2), with each row being 50 m in length and spaced 65 cm apart. Fertilizer was applied once on May 12. The nitrogen application rate was 200 kg N ha^−1^ for maize and 80 kg N ha^−1^ for soybean. Phosphate was applied as heavy superphosphate (P_2_O_5_, 46%) for maize and soybean, at a rate of 120 kg P ha^−1^. Potassium was applied as potassium sulfate (K_2_O, 50%) at a rate of 90 kg K ha^−1^ for maize and soybean. Before seedlings emerged, Acetochlor (C_14_H_20_ClNO_2_, 50%, JINQIU, Changchun, China) was manually sprayed at 1200 mL ha^−1^ during sunny weather from 3 to 5 May each year. In addition, maize was manually weeded twice during the 6- to 12-leaf stage, and maize was manually sprayed with Chlorantraniliprole (C_18_H_14_BrCl_2_N_5_O_2_, 4.5%, JINQIU, Changchun, China) at the jointing stage (during sunny weather from 10 to 15 June each year) at 350 mL ha^−1^. Soybean herbicide and insecticide rates and timing were consistent with maize. At the end of each growing season, maize straw was crushed, plowed, and returned to the field. All other field management methods were consistent across each treatment during the growth period.

### 2.3. Sample Collection

Maize plants were sampled at maturity each year from 2019 to 2023. First, the plant height, ear height, and stem diameter of 10 maize plants in each treatment were measured. The ear position coefficient (%) was calculated as: ear height (cm)/plant height (cm) × 100%. Then, the maize in a 20 m^2^ area in each treatment was harvested. All the grain was threshed, and the kernel weight was measured and converted to yield per hectare based on the area harvested. In addition, we measured ear length, ear diameter, grain number per ear, bald tip length and 100-grain weight. The seed moisture content was determined using a PM8188 (PM-8188-A, KETT, Tokyo, Japan) moisture meter with three replications, and the final yield data were determined using a seed moisture content of 14%.

Maize rhizosphere soil was sampled in 2023. Five soil samples were collected from each plot. Five plants were removed from the ground, and large pieces of soil were removed with a sterile brush so that only about 1 mm of soil was still attached to the roots. The roots were placed in a sterile tube, 50 mL of 1× PBS buffer was added, and the roots were stirred vigorously with sterile forceps for 5 min. Roots were removed and centrifuged at 13,000× *g* for 30 s (TGL-16 s, SHUKE, Chengdu, China). The centrifugal precipitate was considered to be rhizosphere soil and was stored at −80 °C until DNA extraction.

### 2.4. DNA Extraction and Sequencing

Genomic DNA was extracted from rhizosphere soil samples using the E.Z.N.A. Soil DNA Kit (Omega Bio-tek, Inc., Norcross, GA, USA) and stored at −20 °C for subsequent experiments. A 30 ng DNA sample was taken for PCR amplification. Specific primers with barcodes were used to amplify DNA with an ABI 9700 PCR instrument (Applied Biosystems, Inc., Foster City, CA, USA). The quality of DNA extraction was assessed by 1% agarose gel electrophoresis. The ITS1F (5′-CTTGGTCATTTAGAGGAAGTAA-3′) and ITS2 (5′-GCTGCGTTTCTTTCATCGATGC-3′) primers were used to amplify the ITS1 region. The amplification system and procedures are detailed in Table 1.

The library construction comprised the following steps: (1) ligation of the “Y” junction; (2) removal of junction self-associated fragments by magnetic bead screening; (3) enrichment of the library template by PCR amplification; (4) denaturation by sodium hydroxide to produce single-stranded DNA fragments [16]. Sequencing methods and hardware and software information are shown in Table 2. All sequence data were uploaded to the National Center for Biotechnology Information (NCBI; https://www.ncbi.nlm.nih.gov/) (URL, accessed on 1 August 2024) under accession number: PRJNA1123432.

### 2.5. Statistical Analysis

The following statistical analyses were performed using SPSS 22.0 (https://www.ibm.com/spss) (accessed on 6 August 2024) (Norman H. Nie, C. Hadlai Hull and Dale H. Ben, Stanford University, Stanford, CA, USA) software: one-way ANOVA (https://www.ibm.com/spss) (accessed on 6 August 2024) was used to compare the effects of cropping system on maize agronomic traits and yield component factors. After evaluating the significant differences between the sample means via one-way ANOVA, Duncan’s test, which is a post hoc test, was used to determine which specific group means were significantly different from each other. Pearson’s correlation was used to evaluate the relationships between the relative abundance of dominant genera, soil chemical properties, and yield. Beta diversity was determined using the coefficient of variation of the Aitchison distance and was plotted using non-metric multidimensional scaling (NMDS) to compare the degree of similarity between samples.

A soil fungal genera-level network was constructed using the Hmisc package (5.1-1, Frank Harrell, University of Alabama at Birmingham: Birmingham, AL, USA) and the igraph package (2.0.3, Kirill Müller, Karlsruhe Institute of Technology: Karlsruhe, DE, USA) in R version 4.3.1 (MathSoft, 2023; https://www.r-project.org/, accessed on 14 July 2024). The network was analyzed using Gephi 0.9.2 (Netbeans, 2022) [17] and plotted using Java (SE 8u 171, Netbeans, 2022). The model was constructed using R based on the following equations:(1)$$Cb\left(n\right)=\sum_{s\neq n\neq t}\left(\frac{\sigma_{st}\left(n\right)}{\sigma_{st}}\right)$$
(2)$$T_{n}=\frac{avg\left(J_{\left(n,m\right)}\right)}{k_{n}}$$
where *n* is the destination node, *s* and *t* are nodes in the network other than *n*, $\sigma_{st}$ represents the number of shortest paths from node s to node t, and $\sigma_{st}(n)$ denotes the number of shortest paths from node s to node *t* that must pass through node *n*. $J_{\left(n,m\right)}$ is the number of all nodes adjacent to both nodes *n* and *m*, and the value of $J_{\left(n,m\right)}$ is increased by 1 if *n* is directly adjacent to m. $k_{n}$ is the number of all neighbours of the node.

Structural equation modeling (SEM) of the effects of different cropping modes on maize yield pathways was implemented using the ‘Lavaan’ package (0.6–18, Yves Rosseel, Ghent University, BEL) in R v4.3.1 (https://lavaan.ugent.be, accessed on 1 July 2024).

## 3. Results

### 3.1. Maize Plant Growth

MSR and MSI had lower maize ear position coefficients than CMC in 2019–2023 (Figure 3), which indicated a better growth structure and higher lodging resistance. In 2019–2020, the plant height and ear height under CMC were lower than those under MSR and MSI, by 4.21–13.97% and 6.75–8.98%, respectively. The ear position coefficients under CMC were significantly lower than that under MSR, by 4.94–7.6%. In 2021, the plant height under MSR was 4.88–8.52% higher than that under MSI and CMC, and the ear height under MSR was 10.85% higher than that under CMC. However, in 2022–2023, plant height followed the order CMC > MSI > MSR, with the plant height under CMC being 3.31–13.01% higher than that under MSI and MSR, and the plant height under MSI being 9.39% higher than that under MSR. In 2022–2023, the ear position coefficient under MSR was significantly lower than that under MSI and CMC, by 12.96–16.26%.

### 3.2. Maize Yield and Yield Components

The effects of cropping system on maize yield components are shown in Figure 4. Maize ear length and ear diameter varied considerably under CMC from 2019 to 2023 and were lowest in 2022, when they were 9.58% and 11.33% lower than those under MSI and MSR, respectively. From 2019 to 2023, the ear diameter, ear length, and number of grains per ear were similar between MSI and MSR (Figure 4a–c), but in general, they were higher under MSR than under MSI. From 2019 to 2023, the maize bald tip length under each cropping system showed a downward trend (Figure 4d), with the average maize bald tip length decreasing from 0.60 cm in 2019 to 0.06 cm in 2023. However, the maize bald tip length was always higher under CMC than under MSI and MSR, and the 100-grain weight was higher under MSR than under CMC and MSI, by an average of 3.07% (Figure 4e). Overall, the maize yield in each year followed the order MSR > MSI > CMC (Figure 4f). Except for in 2020, the maize yield under MSR generally increased over time. The maize yield under MSI stayed similar over time, with the difference between the highest and lowest yield year being only 1259.7 kg ha^−1^. The maize yield under CMC decreased with time, and the maize yield in 2020–2023 was 5.03–10.99% lower than that in 2019. In summary, compared with CMC, MSR and MSI generally increased maize ear length, ear diameter, and number of grains per ear, reduced maize bald tip length, and thus increased maize yield. Moreover, MSR resulted in a higher maize yield than MSI.

### 3.3. Rhizosphere Soil Fungal Community

#### 3.3.1. Alpha Diversity of Maize Rhizosphere Soil Fungal Community

To investigate soil fungal alpha diversity under different cropping systems, two soil fungal community richness indices (Chao1 and observed number of species; Figure 5a,b) and two community diversity indices (Faith’s phylogenetic diversity (PD) and Shannon diversity; Figure 5c,d) were calculated. The Chao1 and number of observed species under MSR and CMC were higher than those under MSI. The Chao1 and number of observed species under MSR were 48.30% and 29.48% higher, respectively, compared with those under MSI, and the Chao1 and number of observed species under CMC were 62.58% and 40.07% higher, respectively, compared with those under MSI. Faith’s PD under CMC was 14.79% and 10.43% higher compared to that under MSI and MSR, respectively. Shannon diversity under CMC was significantly higher than that under MSI, by 10.07%. In summary, compared with under CMC, the richness and diversity of soil fungal communities under MSI and MSR were lower, with MSI having lower richness and diversity than MSR.

#### 3.3.2. NMDS of Maize Rhizosphere Soil Fungal Community

NMDS was used to explore differences in soil fungal communities among cropping systems (Figure 6). Each cropping system was separated along NMDS2, indicating different fungal community compositions among the cropping systems. MSI and MSR were close to each other in the NMDS plot and far from CMC, indicating that intercropping and rotation systems affected the fungal community structure.

#### 3.3.3. Fungal Community Composition in Maize Rhizosphere Soil under Different CropPing Systems

In total, 707 genera of fungi were annotated. The unnamed genera were removed and the genera with a relative abundance greater than 0.1% were retained, resulting in 18 genera (Figure 7). For MSI and CMC, *Tausonia*, *Ceratobasidium*, *Chaetomium*, and *Mycena* were the dominant genera, accounting for 37.21% and 27.47% of the total relative abundance, respectively. For MSR, the dominant genera were *Tausonia*, *Chaetomium*, and *Solicoccozyma*, accounting for 48.12% of the total relative abundance. Other genera included *Mortierella*, *Crocicreas*, and *Exophiala*. Further analysis showed that the relative abundance of *Tausonia* under MSR was 147.78% and 89.89% higher than that under MSI and CMC, respectively, and the differences were significant. The relative abundance of *Ceratobasidium* under MSI was higher than that under MSR and CMC, by 378.41% compared with CMC. The relative abundance of *Chaetomium* under MSI and MSR was 37.76% and 75.20% higher than that under CMC, respectively. Similarly, the relative abundance of *Mortierella* under MSI and MSR was 40.54% and 47.11% higher compared with that under CMC, respectively. The relative abundance of *Solicoccozyma* was 3.46% under CMC, 1.63% under MSI, and 0.33% under MSR.

#### 3.3.4. Co-Occurrence Network of the Maize Rhizosphere Soil Fungal Community

To determine relationships among genera, we constructed a co-occurrence network of the top 200 fungal genera in maize rhizosphere soil under different cropping systems (Figure 8). Topological parameters of the network are shown in Table 3. All genera in the three cropping systems were from Ascomycota, Basidiomycota, and Mortierellomycota. Under the same network node, there were 17.50% and 3.40% fewer edges under MSI and MSR, respectively, compared with under CMC. In addition, the proportion of positive correlations was 2.01% and 24.92% lower, respectively. The average degree, average weighted degree, average clustering coefficient, and modularity under CMC were higher than those under MSI and MSR. The CMC co-occurrence network had more connections between network nodes, and the connections were stronger and more complex. These results indicate that the rotation and intercropping systems can reduce the complexity of the fungal community network, weakening the positive relationships among genera and increasing competition. This could in turn inhibit pathogens.

#### 3.3.5. Prediction of Fungal Function in Rhizosphere Soil

A total of seven metabolic pathways were annotated across the treatments (Figure 9a–c). The saprotroph pathway was dominant across treatments, ranging from 57.15% to 79.78%. The saprotroph pathway was 30.59% and 37.01% more abundant under MSR than under MSI and CMC, respectively. Other fungal pathways showed the opposite trend (CMC > MSI > MSR). To explore the fungal function further, different functional guilds were annotated (Figure 9d). The average proportion of endophytes under MSR was 22.32–54.13% lower than that under MSI and CMC, while the average proportion of wood saprotrophs was 30.54–57.04% higher. The average proportion of plant pathogens under CMC was 37.18% and 139.53% higher than that under MSI and MSR, respectively.

#### 3.3.6. Effects of Cropping System on Fungal Community Structure and Function and Maize Yield

We used a structural equation model (SEM) to describe the relationship between rhizosphere soil fungal community diversity (FCD) and richness (FCR), proportion of plant pathogens (PP), and maize yield under different cropping systems (Figure 10a). The loading coefficients of the direct effects of CMC, MSI, and MSR on crop yield were −0.42, 0.39, and 0.48, respectively, indicating that MSI and MSR were positively correlated with crop yield, and CMC was negatively correlated with crop yield. Both MSI and MSR reduced FCD, FCR, and PP, but the negative correlation load coefficient of MSR on FCD, FCR, and PP was higher than that of MSI.

Figure 10b shows the results of correlation analysis between the dominant genera of the soil fungal community and maize yield. *Tausonia* and *Mortierella* were significantly positively correlated with maize yield, while *Solicoccozyma* was significantly negatively correlated with maize yield.

## 4. Discussion

Root–shoot relationships are critical for the growth and development of crops and for yield and grain quality [18]. Maize yields are affected by the maize variety and field management, such as the cropping system [19]. Furthermore, yield is a product of different agronomic traits, which also affect each other [20]. For example, a study showed that the ear position coefficient of maize was negatively correlated with lodging resistance, while the stem diameter was positively correlated with lodging resistance and yield per plant [21]. In our study, MSR and MSI resulted in a lower maize ear position coefficient (4.94–7.6%) compared with CMC. This could indicate a healthier root–shoot relationship and growth structure, and likely improved the crop’s resistance to stunting [22]. In addition, both MSR and MSI resulted in smaller maize bald tip lengths and larger maize ear lengths, ear diameters, number of grains per ear, and yield. This could be because crop rotations can change the soil microbial diversity and its ecological function, which in turn affect aboveground growth, development, and yield [23]. Intercropping affects soil microecology through the exchanges and interactions between root systems, affecting the morphology and function of the root system [24]. Intercropping also affects the microclimate of the canopy: depending on the plant types included in the system and the population structure, intercropping affects photosynthesis production, assimilate allocation, and, consequently, yield [25].

Fungi are important decomposers and contribute to soil nutrient and energy cycling [26]. A study using phospholipid fatty acid (PLFA) analysis to compare continuous maize cropping with maize–soybean rotation and intercropping systems found that the soil fungal community was dominated by ascomycetes and that rotation and intercropping significantly reduced soil fungal biomass and increased soil fungal diversity [27]. However, in our study, both MSI and MSR reduced soil fungal diversity. This may be because our study site was established for 11 years (2012–2023). We found that the maize rhizosphere soil fungal communities became more stable over time, with the differences between cropping systems becoming more obvious. Moreover, our co-occurrence network models showed that both MSI and MSR reduced the complexity of the fungal community network, weakening the direct synergistic effects and increasing competition among the fungal genera. These results indicate that crop rotation and intercropping can regulate soil microbial dynamics and reduce the occurrence of soil-borne diseases [28]. The pathotrophic, symbiotrophic, saprotrophic, and pathotrophic–saprotrophic pathways generally dominate fungal communities [29], and in our study, the saprotrophic pathway dominated. However, MSI and MSR decreased the pathotrophic pathway and increased the saprotrophic pathway compared with CMC. Moreover, there were differences in fungal functional guilds. For example, the relative abundance of wood saprotrophs under MSI and MSR was 26.50–57.04% higher. These results indicate that the ability of the fungal community to decompose litter and other substances was enhanced under MSI and MSR. However, these functions were predicted using FUNGuild, and metagenomic technology is needed to improve the accuracy and reliability of these results [30].

In this study, the relative abundance of *Tausonia* and *Mortierella* was higher under both MSI and MSR than under CMC. In addition, both *Tausonia* and *Mortierella* were significantly positively correlated with yield. Of note, the relative abundance of *Tausonia* under MSR was 89.89–147.78% higher than that under MSI and CMC. *Tausonia* is a saprophytic ascomycete that can degrade organic matter, such as keratin and lignin, into nutrients that are readily absorbed by crops [31]. Planting different crops can provide different substrates, such as crop residues and deadwood, for *Tausonia* [32]. This could also explain the high relative abundance of wood saprotrophs under MSR in this study. *Mortierella* are beneficial soil fungi, which can replenish nitrogen and solubilize phosphorus, promote plant growth, and improve plant disease resistance [33]. The secretion of sugars and amino acids by the maize root system promotes microbial decomposition of hard-to-utilize phosphorus, which in turn increases the abundance of *Solicoccozyma* [34]. In our study, the relative abundance of *Solicoccozyma* under CMC was two times that under MSI and 10 times that under CMC. *Solicoccozyma* is a member of the phylum Ascomycota, which is a large and complex group that contains many phytopathogenic fungi such as *Fusarium spinosum* and *Fusarium xylophilum*, both of which cause root rot in soybeans [35]. An increase in the abundance of *Solicoccozyma* can indicate an increase in the incidence of crop diseases [36]. The high abundance of *Solicoccozyma* could explain the high proportion of plant pathogens under CMC in this study.

The application of microbial inoculum has been receiving more and more attention [23]. The beneficial fungi derived from this study could be combined with beneficial bacteria to form an inoculum that could improve soil function, such as nitrogen fixation. Inoculum application could be combined with suitable cropping patterns to increase yields and guarantee the sustainable use of arable land in the black soil zone.

The aim of this study was to compare changes in the structure and function of soil fungal communities under different cropping patterns and, in turn, clarify the effects of these changes on aboveground crop growth. However, the study had several limitations. The experimental sites in this study were limited to the northeastern region of China, and the applicability of these findings to other regions due to differences in climatic conditions, farming systems, soil types, and crop varieties has not been verified. In addition, the crop types in this study were limited to maize and soybean, and the findings may not be applicable to other crop combinations in rotation and intercropping patterns. Finally, there are differences in agricultural management between dryland and paddy fields, and soil microbial communities are closely related to water and fertilizer management, so the results of this study are not applicable to paddy fields. Therefore, in future studies, further research across multiple regions and diverse crop combinations is needed to gain a more comprehensive understanding of their effects on soil microbial communities under different cropping patterns.

## 5. Conclusions

Compared with continuous maize cropping, rotation and intercropping reduced the ear position coefficient, improved the root–shoot relationship, and increased the yield of maize. The increased yield was partly driven by a decrease in the diversity and richness of rhizosphere soil fungal communities. Specifically, the relative abundance of *Tausonia* and *Mortierella*, which are beneficial fungi, increased while the relative abundance of *Solicoccozyma*, a harmful fungus, decreased. Overall, the proportion of wood saprotrophs increased, and the proportion of plant pathogens decreased. These results indicate that intercropping and crop rotation improved soil fertility and decreased the occurrence of soil-borne diseases. In summary, this study concluded that both intercropping and crop rotation are effective ways to increase crop productivity and improve the soil microenvironment. Therefore, a combination of crop rotation and intercropping into a composite cropping system should be considered. For example, the new model could include alternate intercropping characterized by intercropping of two crops with inter-annual swapping of planting positions. This could combine the advantages of intercropping and crop rotation to improve root–shoot relationships and promote crop yield and quality for sustainable agricultural development.

## Figures and Tables

**Figure 1 microorganisms-12-01620-f001:**
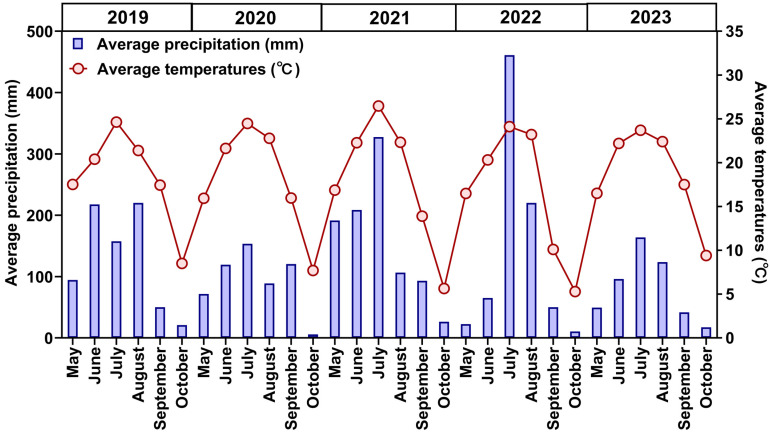
Monthly average rainfall and average temperature during the sampling period (2019–2023).

**Figure 2 microorganisms-12-01620-f002:**
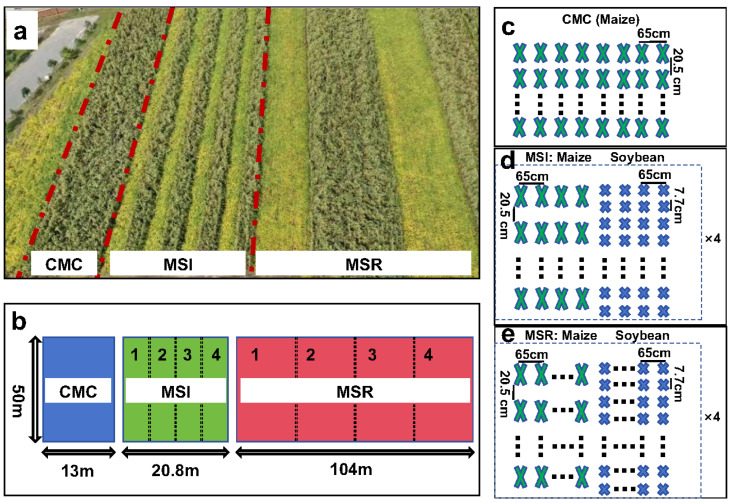
Experimental design. (**a**) Aerial photograph of the long-term field experiment; (**b**) diagram of the field distribution of the three cropping patterns; 1–4 indicate replicate plots; (**c**–**e**) diagram of the distribution of crops planted in each treatment; 65 cm refers to the space between rows, while 20.5 cm and 7.7 cm refer to the space between plants within a given row. Continuous maize cropping (CMC), maize–soybean rotation (MSR), maize–soybean intercropping (MSI).

**Figure 3 microorganisms-12-01620-f003:**
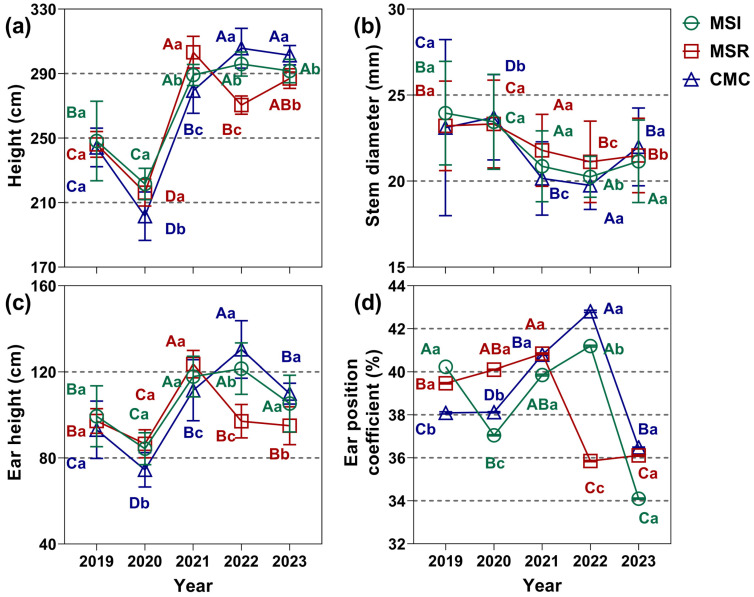
Effects of cropping system on maize growth in 2019–2023. (**a**) Plant height, (**b**) stem diameter, (**c**) ear height, and (**d**) ear position coefficient. Values are means ± SE, n = 10 replicates. The lowercase letters (a–c) indicate differences among cropping systems within the same year, while the uppercase letters (A–D) indicate differences among years under the same cropping system (*p* < 0.05). MSI, maize–soybean intercropping; MSR, maize–soybean rotation; CMC, continuous maize cropping.

**Figure 4 microorganisms-12-01620-f004:**
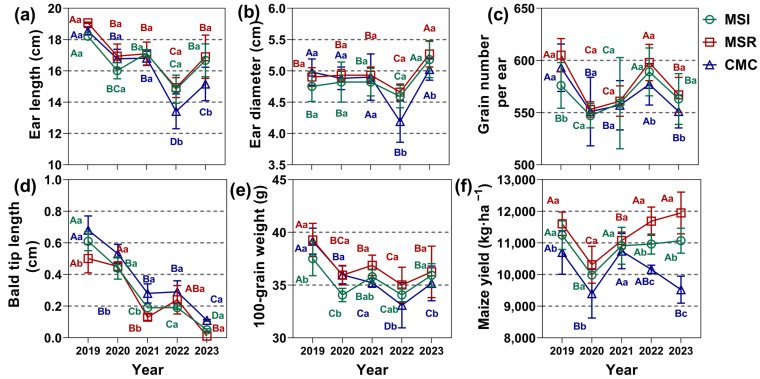
Effects of cropping system on maize yield and yield components in 2019–2023. (**a**) Ear length, (**b**) ear diameter, (**c**) grain number per ear, (**d**) bald tip length, (**e**) 100-grain weight, and (**f**) yield. Values are mean ± SE, n = 10 replicates. The lowercase letters (a–c) indicate differences among cropping systems within the same year, while the uppercase letters (A–D) indicate differences among years under the same cropping pattern (*p* < 0.05). MSI, maize–soybean intercropping; MSR, maize–soybean rotation; CMC, continuous maize cropping.

**Figure 5 microorganisms-12-01620-f005:**
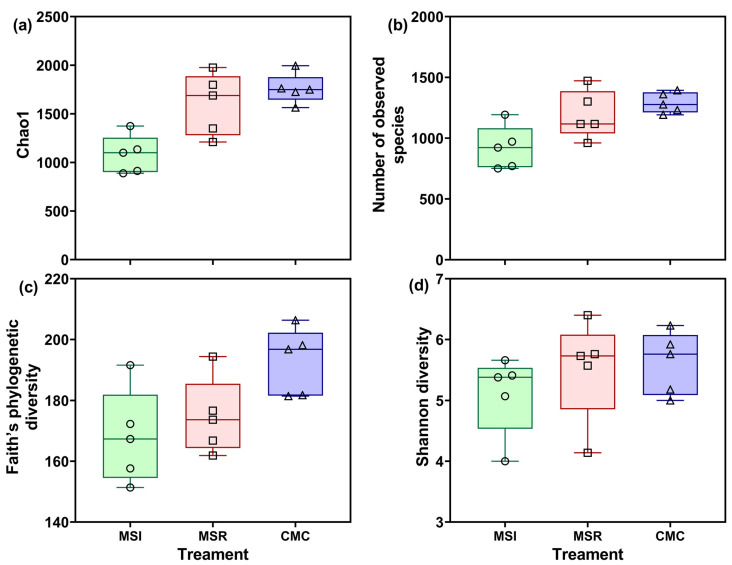
Effects of cropping system on maize rhizosphere soil fungal richness and diversity. (**a**) Chao1, (**b**) number of observed species, (**c**) Faith’s phylogenetic diversity (PD), and (**d**) Shannon diversity. The two ends of the box plot are the upper and lower quartiles, the horizontal line in the middle indicates the median, and the lines connecting the two ends are the minimum and maximum values (*n* = 5). MSI, maize–soybean intercropping; MSR, maize–soybean rotation; CMC, continuous maize cropping.

**Figure 6 microorganisms-12-01620-f006:**
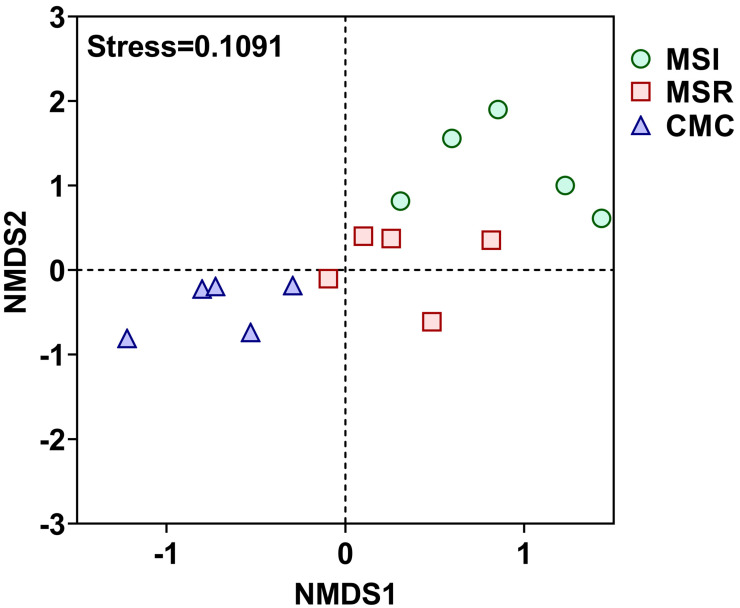
NMDS plot of maize rhizosphere soil fungal communities under different cropping systems. Stress = 0.1091. MSI, maize–soybean intercropping; MSR, maize–soybean rotation; CMC, continuous maize cropping.

**Figure 7 microorganisms-12-01620-f007:**
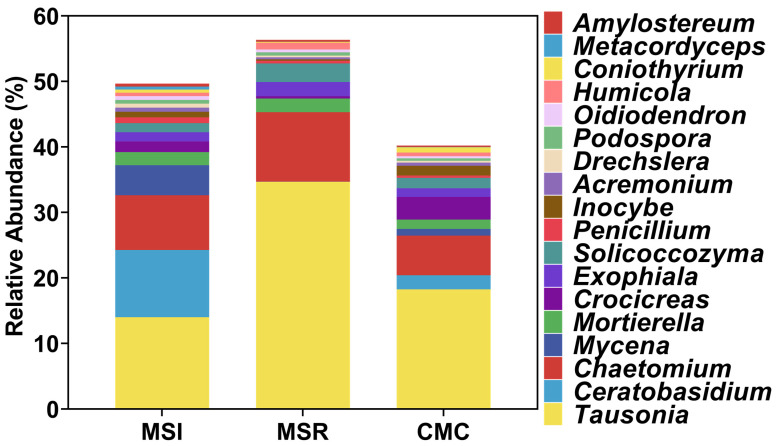
Composition of fungal genera in maize rhizosphere soil under different cropping systems. MSI, maize–soybean intercropping; MSR, maize–soybean rotation; CMC, continuous maize cropping.

**Figure 8 microorganisms-12-01620-f008:**
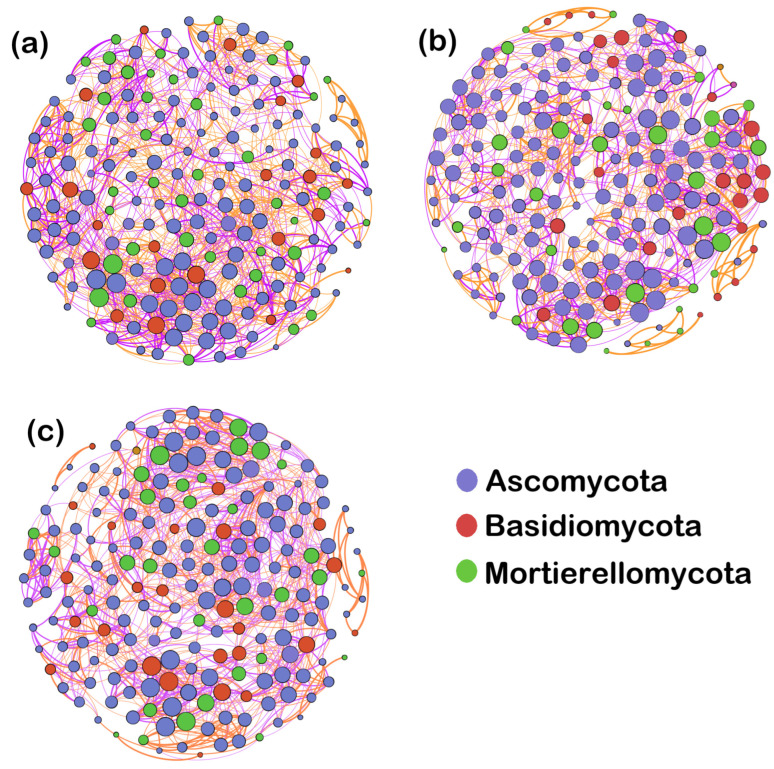
Co-occurrence networks of maize rhizosphere soil fungi under different cropping systems. (**a**) Maize–soybean intercropping, (**b**) maize–soybean rotation, and (**c**) continuous maize cropping. Circles indicate different genera. Genera from the same phylum are labeled with the same color. The size of the circles indicates the average abundance of the genera, lines indicate correlations between two genera, and the line thickness indicates the strength of the correlation. Orange lines indicate positive correlations and purple lines indicate negative correlations.

**Figure 9 microorganisms-12-01620-f009:**
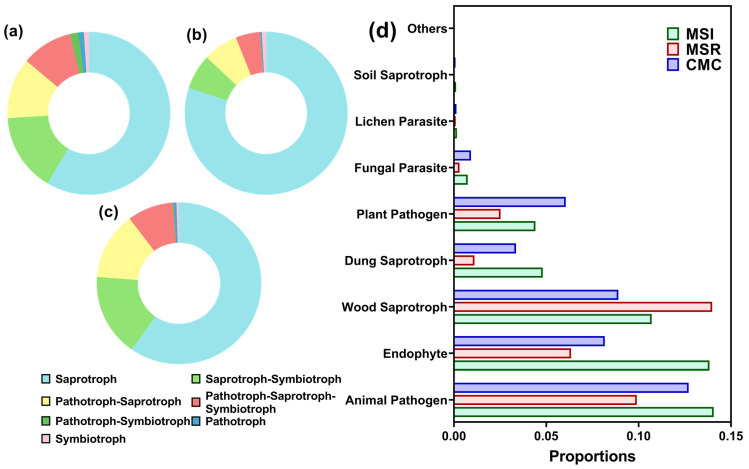
Function of maize rhizosphere soil fungi under different cropping patterns predicted using FUNGuild. (**a**–**c**) Primary metabolic pathways ((**a**) MSI; (**b**) MSR; (**c**) CMC)) and (**d**) secondary metabolic pathways. MSI, maize–soybean intercropping; MSR, maize–soybean rotation; CMC, continuous maize cropping.

**Figure 10 microorganisms-12-01620-f010:**
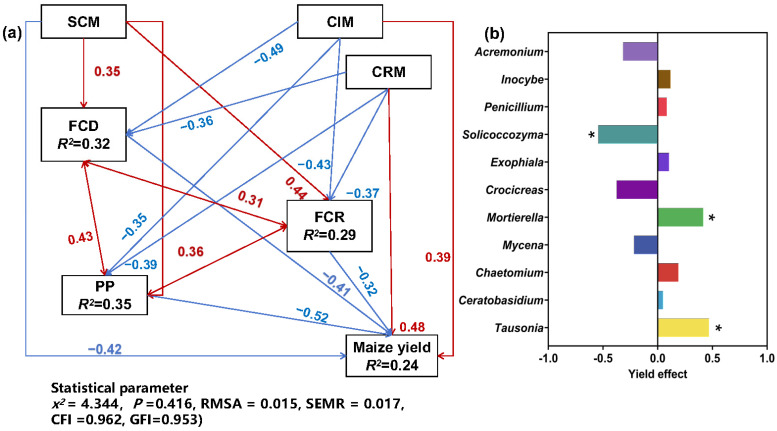
Effects of cropping systems and fungal community structure and function on maize yield. (**a**) Structural equation model (SEM) of the effects of different cropping systems on the structure and function of the fungal community in maize rhizosphere soil and on yield. Blue arrows indicate negative correlations and red arrows indicate positive correlations between variables. MSI, maize–soybean intercropping; MSR, maize–soybean rotation; CMC, continuous maize cropping; FCD, fungal community diversity; FCR, fungal community richness; PP, relative abundance of plant pathogens. (**b**) Correlation analysis between dominant soil fungal genera and maize yield. * *p* < 0.05.

**Table 1 microorganisms-12-01620-t001:** PCR amplification system and procedure for fungi in soil.

PCR Amplification System		PCR Amplification Procedure	
Reagent Composition	Volume	Reaction	
DNA template	30 ng	① Pre-denaturation	94 °C 2 min
Forward Primer (5 uM)	1 μL	② Denaturation	94 °C 15 s
Reverse Primer (5 uM)	1 μL	③ Anneal	55 °C 30 s
BSA (2 ng μL^−1^)	3 μL	④ Extension	72 °C 60 s
2x Taq Plus Master Mix	12.5 μL	⑤ Final extension	72 °C 7 min
ddH_2_O	7.5 μL	③–⑤ Number of cycles	30
Total	25 μL	⑥ Reaction termination	10 °C ∞

**Table 2 microorganisms-12-01620-t002:** Statistics of sequencing instruments and reagents.

Step	Instrument/Reagents	Manufacturer	Specification/Model/Lot Number
Amplicon extraction	MoBio PowerSoil DNA Isolation Kit (100)	QIAGEN (Frankfurt, Germany)	100 times
Amplifier amplification	KAPA 2G Robust Hot Start Ready Mix	KAPA (Boston, MA, USA)	
	ABI 9700 PCR	ABI (Guangzhou, China)	
Amplicon purification	Agencourt^®^ AMPure^®^ XP	Beckman Coulter (Shanghai, China)	Dispense 45 mL/bottle, total 450 mL/bottle
Amplicon building	NEBNext Ultra II DNA Library Prep Kit	NEB (Beijing, China)	96 reactions
	Agencourt^®^ AMPure^®^ XP	Beckman Coulter	Dispense 45 mL/bottle, total 450 mL/bottle
	ABI 9700 PCR	ABI	
Library quality control (instruments)	Bioanalyzer (Agilent 2100)	Agilent (Palo Alto, CA, USA)	DE13806339
	Biomolecule Analyzer (Labchip GX)	PerkinElmer (Shanghai, China)	
	ABI Qpcr	ABI	Step One Plus (Shanghai, China) (www.PuDi.cn) (accessed on 1 August 2024)
Library quality control (reagents)	Agilent DNA 1000 Kit	Agilent (Palo Alto, CA, USA)	300 samples
	HT DNA-Extended Range LabChip	PerkinElmer (Shanghai, China)	
	KAPA Library Quantification Kit	KAPA	500 times
Sequencing (equipment)	High-throughput second-generation sequencer	Illumina (Beijing, China)	MiSeq (Shanghai, China) (www.PuDi.cn) (accessed on 1 August 2024)
Sequencing (reagents)	MiSeq^®^ Reagent Kit v3 (600 cycle) (PE300)	Illumina	
	MiSeq Reagent Kit v2 (500 cycle)	Illumina	

**Table 3 microorganisms-12-01620-t003:** Topological characteristics of maize rhizosphere soil fungal co-occurrence networks.

Treatment	Node	Edge	Positive (%)	Negative (%)	Average Degree	Average Weighting	Modularity	Cluster Coefficient
MSI	201	1114	57.95	42.05	11.14	10.36	0.68	0.65
MSR	199	1266	47.32	52.68	12.74	11.77	0.65	0.62
CMC	200	1309	59.11	40.89	13.03	12.04	0.63	0.61

Note: MSI, maize–soybean intercropping; MSR, maize–soybean rotation; and CMC, continuous maize cropping.

## Data Availability

The original contributions presented in the study are included in the article, further inquiries can be directed to the corresponding authors.

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
