# Peer review of "Maize–Soybean Rotation and Intercropping Increase Maize Yield by Influencing the Structure and Function of Rhizosphere Soil Fungal Communities"

_microorganisms, 2024, doi:10.3390/microorganisms12081620_

Round 1

Reviewer 1 Report

Comments and Suggestions for Authors

I ask the Authors to indicate the possible practical application of the research results. What research problems still need to be solved?

Comments

Line 70, please include a figure with the location of the study sites.

Line 72, Figure 1?

Please complete the manuscript with a complete characterization of soil conditions in each year of the study. Please provide multi-year data. It is necessary to include the Sielianinov hydrothermal index. Please include the dates of agrotechnical treatments. Were crop protection products used in the experiment? If this, what kind and when?

Line 8 , please provide a brief description of the tested maize and soybean varieties

Line 91, 92, doses of phosphorus and potassium please provide in elemental form (P and K)

Line 158, please provide full details of the producer of the statistical software.

Reviewer 2 Report

Comments and Suggestions for Authors

Dear Authors,

Correction suggestions are described in the attached file.

Best regards,

Comments on the Quality of English Language

Dear Authors,

Correction suggestions are described in the attached file.

Best regards,
